# Correlates of low birth weight and preterm birth in India

**Arup Jana** [ID]*

Department of Population & Development, International Institute for Population Sciences, Mumbai, Maharashtra, India

* arupjana0000@gmail.com

**Data Availability Statement:** The study utilized the secondary data, which is publicly available though https://dhsprogram.com/methodology/survey/survey-display-541.cfm.

**Funding:** The authors received no specific funding for this work.

## Abstract

### Background

In the 21st century, India is still struggling to reduce the burden of malnutrition and child mortality, which is much higher than the neighbouring countries such as Nepal and Shri Lanka. Preterm birth (PTB) and low birth weight (LBW) predispose early-age growth faltering and premature mortality among children below the age of five. Thus, highlighting the determinants of LBW and PTB is necessary to achieve sustainable development goals.

### Objective

The present study provides macro-level estimates of PTB and LBW and aims to highlight the nature of the association between various demographic, socioeconomic, and maternal obstetric variables with these outcomes using a nationally representative dataset.

### Methods

Data on 170,253 most recent births from the National Family health survey (NFHS-5) 2019–21 was used for the analysis. The estimates of PTB and LBW are measured by applying sample weights. The correlates of LBW and PTB were analyzed using logistic models.

### Results

There were cross-state disparities in the prevalence of PTB and LBW. In India, an estimated 12% and 18% of children were LBW and PTB, respectively, in 2019–21. Maternal obstetric and anthropometric factors such as lack of antenatal care, previous caesarean delivery, and short-stature mothers were associated positively with adverse birth outcomes such as LBW and PTB. However, a few correlates were found to be differently associated with PTB and LBW. Mothers belonging to richer wealth status had higher chances of having a preterm birth (OR = 1.16, 95% CI: 1.11–1.20) in comparison to poor mothers. In contrast, the odds of having LBW infants were found to be increased with the decreasing level of the mother's education and wealth quintile.

**Competing interests:** The authors have declared that no competing interests exist.

## Conclusions

In India, PTB and LBW can be improved by strengthening existing ante-natal care services and evaluating the effects of the history of caesarean births on future pregnancies.

## Introduction

The adverse birth outcome (ABO) is a complex outcome influenced by multiple factors, encompassing preterm birth, low birth weight, stillbirth, macrosomia, and congenital anomalies. Preterm birth (PTB) is defined as delivery before 37 weeks of gestation, while low birth weight (LBW) refers to infants born weighing less than 2500 g [1, 2]. Stillbirth refers to the delivery of a baby without any signs of life, while macrosomia describes newborns who are larger than the average size. Congenital anomalies are a diverse range of structural or functional abnormalities present at birth, originating during prenatal development [3]. Globally, approximately 15 million premature births [4] and over 20 million infants are born with LBW each year [5]. The prevalence of PTB worldwide is 10.6%, with South Asia accounting for more than one-third of the burden. Additionally, nearly 3 million stillbirths occur annually worldwide, with 98% happening in developing countries [6].

Preterm birth is a significant public health challenge globally, with PTB being the leading cause of neonatal deaths and the second leading cause of under-five mortality [7]. The Sustainable Development Goals (SDGs) set out to reduce preventable deaths in newborns and children under the age of five by 2030, with a goal of reducing neonatal mortality to at least 12 per 1,000 live births and under-five mortality to at least 25 per 1,000 live births [8]. However, in India, PTB remains a pressing issue, with an estimated 3.5 million cases of PTB in 2010, accounting for 23.4% of the global burden [9].

Low birth weight is also a significant issue, with an estimated 15% to 20% of all births worldwide being LBW and the highest prevalence in South Asian countries, reaching up to 28% [10]. In rural India, the overall prevalence of ABOs is more than 25%, adding to the public health crisis [11]. Although the prevalence of LBW in India has decreased from 21% to 18% in the last decade, the country may still face challenges in achieving the child health and mortality targets set by the SDGs [12]. In 2020, the Sample Registration System (SRS) report indicated a rate of 3.8 per 1000 infants born without life [13]. The prevalence of LBW and PTB in India, at 18% and 13%, respectively, is higher than in neighbouring countries such as Sri Lanka (LBW:15.9 and PTB: 7.0), Nepal (LBW:12.3 and PTB: 5.3), Myanmar (LBW:8.1 and PTB: 10.4), and China (LBW:6.9 and PTB: 0) [14]. These findings emphasize the urgent need for improved interventions and strategies to address the prevalence of PTB and LBW in India and prevent neonatal deaths and under-five mortality.

Implementing active policies and programs to manage and decrease the prevalence of PTB and LBW can not only directly reduce the burden of preventable under-five mortality [15]. Still, it can also significantly enhance the human capital stock and overall human development [16]. Adverse birth outcomes, including PTB and LBW, are strong predictors of growth faltering and nutrition deficits in children [17, 18], as well as physical and cognitive development in later life [19]. Consequently, addressing these adverse birth outcomes remains a crucial policy concern from both a public health perspective and for the quality of future human capital in the nation.

A variety of determinants influence adverse birth outcomes. These can include maternal factors, such as maternal age, race/ethnicity, and pre-existing health conditions (e.g., diabetes, hypertension) [20, 21]; lifestyle factors, such as tobacco, alcohol, and drug use during

pregnancy; inadequate prenatal care; and social determinants of health, such as socioeconomic status, education level, access to healthcare, and environmental factors (e.g., exposure to pollution, neighbourhood safety) [22, 23]. Additionally, genetics, maternal stress, and intergenerational factors can also play a role in determining birth outcomes [24]. A complex interplay of these determinants can impact fetal development and increase the risk of adverse birth outcomes, highlighting the importance of addressing multifactorial influences to improve maternal and infant health outcomes.

There is a lack of consensus in research on the underlying causes of adverse birth outcomes, such as PTB and LBW. Thus, this study focused on LBW and PTB as adverse birth outcomes. A hospital-based study in South Africa revealed that inadequate antenatal care, previous caesarean section, and hypertensive disorders had a negative impact on birth outcomes [25]. Maternal anthropometric measures and nutrition outcomes are significantly related to both LBW and PTB [26]. Maternal education, wealth status, and infant birth order were also found to contribute significantly to LBW [27]. Studies based on the Indian population have shown that anaemic mothers in high-risk age groups (below 19 years or above 40 years), lack of proper antenatal care, and inadequate intake of Iron Folic Acid (IFA) supplements during pregnancy are associated with a higher risk of PTB and small-for-gestational-age births [5, 28]. Additionally, maternal or neonatal infectious diseases pre-delivery, hypertension, pre-labour caesarean delivery, and limited accessibility and healthcare availability also negatively affect PTB in India [29, 30]. However, most studies on PTB and LBW in India were hospital-based or small-scale primary studies, and thus, they failed to provide nationally representative estimates.

In recent years, India has significantly reduced under-five mortality [31]. However, it still has a high child mortality rate compared to other mortality [32]. Various programs and policies have been implemented to prevent neonatal and child mortality, such as promoting optimal antenatal checks, institutional deliveries, skilled birth attendants, kangaroo care, postnatal check-ups for mother and child, early initiation of breastfeeding, exclusive breastfeeding, age-appropriate complementary feeding, and immunization [33, 34]. Despite these efforts, preterm birth and LBW continue to be significant contributors to neonatal and child mortality in India, with 3.5 million babies born prematurely and 0.3 million children dying each year before the age of 5 due to complications from these adverse birth outcomes [35]. Most of the research conducted on this issue is based on small sample sizes from hospitals, highlighting the need for sub-national estimates of PTB and LBW using large-scale data. Therefore, this study aims to use data from the Indian National Family and Health Survey (NFHS), round 5, 2019–21, to provide sub-national estimates of PTB and LBW and examine their correlates based on individual-level medical history, demographic, social, and economic backgrounds of families. This study can provide in-depth and comprehensive knowledge of the factors associated with adverse birth outcomes, which can help India in addressing issues like nutrition failures in the early stages of life and premature mortality.

## Data and methods

### Data

The present study utilizes data on LBW and preterm birth in India from the fifth round of the National Family Health Survey (NFHS 5, 2019–21). NFHS5 was conducted across 36 states and union territories and provided information on various important Maternal and Child Health (MCH) indicators. The survey's first round was conducted in 1992–93, and since then, there have been four consecutive rounds, providing cross-sectional data using a multistage cluster sampling design [12]. Further details on the sampling method can be

obtained in the Indian national report, available at http://rchiips.org/NFHS/NFHS-4Report. shtml.

The NFHS-5 sample was designed to provide accurate estimates of all key indicators at the national, state, and district levels (for all 707 districts in India as of March 31, 2017). To achieve this, a sample size of around 636,699 households was determined. The primary sampling units (PSUs) were villages selected with probability proportionate to size, and the rural sample was selected using a two-stage sample design. In the second stage, 22 households were randomly chosen from each PSU. A two-stage sample design was also used in urban areas, with 22 households selected in each Census Enumeration Block (CEB) in the second stage. After mapping and listing households in the selected first-stage units, households were selected for the second stage in both urban and rural areas. The survey covered 724,115 women in the reproductive age group of 15–49 years, and information about 232,920 children was collected from their mothers. Of these children, a total of 170,253 most recent (i.e., last-born) births were included in the study.

The study is based on publicly available secondary data. It is certified that all applicable institutional and governmental regulations concerning the ethical use of human volunteers were followed during the course of the survey. Verbal and written informed consent was obtained from all participants or their parent/legal guardian if they were not of mature age or below 18 years old.

The study includes information on the birth weight of the last live birth, which is available for analysis. Additionally, preterm birth has been calculated based on information on pregnancy duration available for all births that occurred within the five-year period prior to the survey. The data provides a comprehensive reproductive history of women up to the date of the survey and detailed information on birth outcomes during the reference period. Data on birth weight was collected using two methods: maternal recall and birth cards. Preterm birth was determined based on calendar-format pregnancy duration data for all births that occurred within the reference period.

## Outcome variable

Preterm birth is defined as a live birth before 37 completed weeks of gestation, and low birth weight is defined by the World Health Organization (WHO) as weight at birth less than 2500 grams [36]. Jana et al. (2022) was followed in this study to measure preterm birth using the calendar method of DHS [37]. The NFHS collected data on birth weight using the following questions: Was (name of the child) weighed at birth? How much did (name of the child) weigh? The information was reported in two ways; first, the mother's recall of her baby's weight, and second, reported with the help of any card of their baby's weight [12]. The other outcome variable was preterm birth, estimated using the pregnancy duration. The outcome variables are dichotomous, 0 signifying the child did not have LBW/ was not preterm and 1 denoting the child had LBW/ was preterm.

## Exposure variables

The correlates of adverse birth outcomes were selected based on published literature and the availability of variables in the data set. We evaluated the association of maternal obstetrics factors, socioeconomic, demographic and child factors with the birth outcomes. The maternal obstetrics factors included any complication at the time of delivery, the place of delivery of the child, previous caesarean delivery, anaemia of the mother, receiving full antenatal care and the mother's height. The socioeconomic and demographic factors were religion, caste, wealth status, mother's education, type of residence, drinking water sources, and

mother's age at delivery. The child factors were birth order and sex of the child. Full antenatal care (ANC) indicates the mother receiving at least 3+ANC visits, 2 tetanus toxoids (TT) taken, and 100 IFA consumed at the time of pregnancy. The complications during pregnancy were categorized into two categories; "Any complication" (experienced vaginal bleeding, convulsions, prolonged labour, severe abdominal pain and high blood pressure) and "No complication". The place of delivery is coded as a dichotomous variable as "Private health facility" and "Public health facility". History of Caesarean delivery is coded as "Yes" and "No". Mother's general health plays a pivotal role in determining the health of newborns [38]. The study has used a mother's height as an anthropometric measure of healthy growth because of its reliability and stability in adulthood. Mother's height was categorized as "<145 cm.", "145–149", "150–154", and ">155". Further, a binary variable for the mother's anaemia is another proxy for maternal health and nutritional status. Maternal anaemia is documented to have strong associations with poor birth outcomes [39]. Some of the other biological and demographic characteristics controlled for are the mother's age at delivery, categorized into "below 24", "24–29", and "30 & above". Birth parity was controlled through birth order recoded as '1', '2', and '3 & above', and the birth interval was controlled as two categories: "<24 months" and "> = 24 months". The study has also controlled for some Water, Sanitation and Hygiene (WASH) indicators like the source of drinking water and sanitation. The study has controlled for the availability of improved sanitation in the household. The variable is defined as per the NFHS norms [12]. The household's socioeconomic status, such as wealth status, religion, social caste, and place of residence, was also adjusted for the analysis.

## Statistical analysis

The prevalence of preterm birth and LBW was calculated by estimating the number of adverse events in the exposed sample of the last live birth. Association and goodness of fit between adverse birth outcomes and control variables of interest were first determined using the Chi-square statistic ($\chi2$). The bivariate analysis was performed to estimate the prevalence of the outcome variables by selected characteristics. Further, a multivariate logistic regression model has been applied to estimate the association between the selected demographic, health, social and economic correlates with LBW and preterm birth. Three models were used in the present study, as it has previously proven to be helpful in understanding the significance of various groups of correlates [25]. In model 1, maternal obstetric factors were taken. Model 2 consists of demographic characteristics, and Model 3 considers socioeconomic factors. Model 4 is a full model with all control variables. All of the analysis for the present study was done using STATA, version 14.1. and Arc Map version 13.

## Results

### Prevalence of adverse birth outcomes

Approximately 12% of children were born preterm, and 18% had low birth weight in India during 2019–21. Fig 1 displays the state-wise prevalence of preterm birth and LBW. Among all states and UTs, the self-reported prevalence of LBW was highest in Punjab (22%), followed by Delhi, Dadra & Nagar Haveli, Madhya Pradesh, Uttar Pradesh, and Haryana (Fig 1A). The incidence rate of preterm birth was concentrated in states like Himachal Pradesh, Uttarakhand, Andaman & Nicobar Island, Rajasthan, Nagaland, and Delhi (Fig 1B). The prevalence of preterm birth ranges from 39% in Himachal Pradesh to 2% in Mizoram.

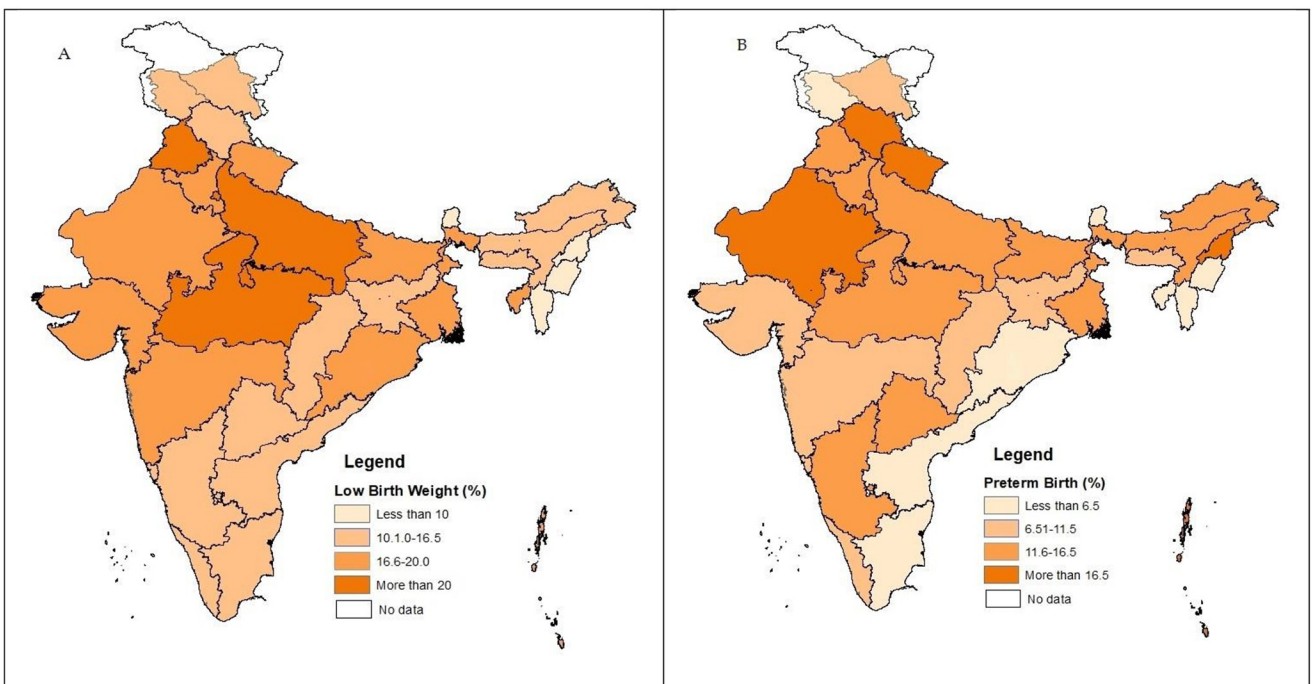

**Fig 1. Spatial pattern of low birth weight (A) and preterm birth (B) in India, 2019–21.** Source: NFHS-5, prepared by the author.

### Risk factors for preterm birth

Table 1 shows that mothers who did not receive antenatal care had 13% preterm. The prevalence of LBW (18%) was higher among mothers who had faced any complications during pregnancy as compared to mothers who did not experience complications. Preterm birth was higher among infants delivered at home and private hospitals (13%) compared to those delivered at public health institutions (12%). On the other hand, LBW was much higher among babies born at home (22%) compared to health facilities (17%). The mothers who had previous caesarean delivery had a higher prevalence of preterm birth (21%) in comparison to those who had vaginal deliveries (18%).

Table 2 presents the odds ratios obtained from logistic regression analysis to assess the association between various factors and PTB. The first model showed that mothers who did not receive full antenatal care during pregnancy had a higher risk of experiencing preterm birth (OR = 1.25). Mothers with previous caesarean delivery were also more likely to have a preterm birth than those without (OR = 1.08, 95% CI: 1.04–1.13). The chance of preterm birth was significantly higher among those who delivered at a private hospital than those who delivered at home (OR = 1.24, 95% CI: 1.17–1.31). Model II examined the effects of infants' and mothers' demographic factors on preterm birth. Mothers in the mature age group (24–29 years) were more likely to have preterm birth compared to young mothers (aged below 24) (OR = 1.07, 95% CI: 1.02–1.11). Female children had a higher probability of being preterm than male children. In model III, the chance of having preterm birth was lower among higher-educated mothers compared to illiterate/primary-educated mothers. Moreover, mothers belonging to richer wealth status had a higher likelihood of having a preterm birth (OR = 1.16, 95% CI: 1.11–1.20) than poor mothers. After adjusting for all plausible factors of preterm birth in the final model, it was found that older mothers (aged 30 years and above) had a lower likelihood

**Table 1. Distribution of preterm birth and low birth weight by background characteristics.**

| Control variables | Total Births | Preterm birth (%) | Low birth weight (%) | χ2 (PTB) | χ2 (PTB) |
|---|---|---|---|---|---|
| **Maternal obstetric care** | | | | | |
| Full antenatal care | | | | 138.35*** | 142.62*** |
| No | 119,479 (70.18) | 13.37 | 18.56 | | |
| Yes | 50,774 (29.82) | 10.44 | 16.13 | | |
| Pregnancy complication | | | | 9.59*** | 20.31*** |
| No | 46,015 (27.03) | 13.26 | 18.24 | | |
| Any complication | 124,238 (72.97) | 12.13 | 17.57 | | |
| Place of delivery | | | | 45.30*** | 85.48*** |
| Home | 20,515 (12.05) | 13.08 | 21.64 | | |
| Public | 111,229 (65.33) | 12.10 | 17.69 | | |
| Private | 38,509 (22.62) | 12.98 | 17.15 | | |
| Previous caesarean delivery | | | | 43.81*** | 10.71*** |
| Yes | 3,265 (1.92) | 12.86 | 18.41 | | |
| No | 22,265 (13.07) | 12.86 | 21.46 | | |
| No preceding birth | 144,730 (85.01) | 12.33 | 17.55 | | |
| Anaemia | | | | 57.99*** | 16.73*** |
| Anaemic | 99,705 (58.56) | 12.24 | 18.13 | | |
| Not anaemic | 70,548 (41.44) | 12.73 | 17.19 | | |
| **Demographic variables** | | | | | |
| Mother's height | | | | 566.83*** | 1.48 |
| <145 | 19,211 (10.28) | 12.49 | 23.30 | | |
| 145–149 | 43,363 (25.47) | 12.84 | 19.77 | | |
| 150–154 | 57,707 (33.89) | 12.39 | 16.68 | | |
| >155 | 49,972 (29.35) | 12.12 | 15.13 | | |
| Mother's age at birth | | | | 222.64*** | 19.78*** |
| <24 | 83,890 (49.27) | 12.75 | 18.86 | | |
| 24–29 | 54,843 (32.21) | 12.00 | 16.51 | | |
| 30 & above | 31,520 (18.51) | 12.28 | 16.49 | | |
| Birth order | | | | 69.13 *** | 17.30 |
| 1 | 57,303 (33.66) | 12.39 | 18.58 | | |
| 2 | 60,094 (35.30) | 12.30 | 16.99 | | |
| 3 & above | 52,856 (31.05) | 12.67 | 17.68 | | |
| Birth interval | | | | 53.24*** | 34.15*** |
| <24 months | 26,737(23.80) | 12.89 | 17.61 | | |
| = >24 months | 85,620 (76.20) | 12.56 | 15.62 | | |
| Sex of child | | | | 175.89*** | 6.52** |
| Male | 91,360 (53.66) | 12.59 | 16.30 | | |
| Female | 78,893 (46.34) | 12.26 | 19.42 | | |
| **Socio-economic status** | | | | | |
| Religion | | | | 286.02*** | 40.92*** |
| Hindu | 125,458 (73.69) | 12.43 | 17.99 | | |
| Non -Hindu | 44,795 (26.31) | 12.46 | 16.73 | | |
| Wealth status | | | | 280.24*** | 52.05*** |
| Poor | 82,768 (48.61) | 12.70 | 20.05 | | |
| Middle | 33,469 (19.66) | 11.91 | 17.14 | | |
| Rich | 54,016 (31.73) | 12.41 | 15.50 | | |

(*Continued*)

**Table 1.** (Continued)

| Control variables | Total Births | Preterm birth (%) | Low birth weight (%) | χ2 (PTB) | χ2 (PTB) |
|---|---|---|---|---|---|
| Mother's education | | | | 398.85*** | 20.53*** |
| Illiterate & primary | 55,583 (32.65) | 13.14 | 20.13 | | |
| Secondary | 89,569 (52.61) | 12.03 | 17.81 | | |
| Higher | 25,101 (14.74) | 12.39 | 13.68 | | |
| Place of residence | | | | 19.38*** | 3.16 * |
| Rural | 134,429 (78.96) | 12.69 | 18.08 | | |
| Urban | 35,824 (21.04) | 11.78 | 16.89 | | |
| Source of drinking water | | | | 0.45 | 3.47* |
| Unimproved | 19,585 (11.50) | 12.45 | 18.18 | | |
| Improved | 150,668 (88.50) | 12.44 | 17.69 | | |
| Sanitation facility | | | | 196.45*** | 0.75 |
| Improved | 121,879 (71.59) | 12.28 | 17.04 | | |
| Unimproved | 48,373 (28.41) | 12.82 | 19.53 | | |
| Region | | | | 751.25*** | 725.52*** |
| North | 32,100 (18.85) | 17.05 | 18.24 | | |
| Central | 41,573 (24.42) | 13.86 | 19.65 | | |
| East | 32,348 (19.00) | 12.42 | 17.50 | | |
| North-East | 26,908 (15.80) | 10.53 | 14.45 | | |
| West | 15,459 (9.08) | 9.19 | 18.73 | | |
| South | 21,865 (12.84) | 9.50 | 15.05 | | |
| India | 170,253 | 12.44 | 17.74 | | |

Note: Percent distribution provided within parenthesis

(OR = 0.88, 95% CI: 0.78–0.98) of having PTB than younger mothers. Additionally, antenatal care and place of delivery were significant factors of PTB.

### Risk factors for low birth weight

Mothers did not receive full antenatal care and had faced complications during pregnancy had LBW children. The percentage of LBW was almost 8 percentage points more (Table 1) among mothers with less than 145cm height than those who had more than 155cm height (15%). Young mothers had a higher percentage of PTB, whereas it was the opposite in the case of LBW. The percentage of LBW decreased with an increase in the mother's height, education, and household wealth status. Among mothers with a history of previous caesarean delivery, almost 22% of children had LBW. The prevalence of LBW babies was found to decrease with the mother's age.

The odds ratios from logistic regression to estimate the association of correlates with LBW in India are presented in Table 3. The results of the first model show that a history of previous caesarean delivery significantly increases the odds of delivering a child with LBW (OR = 1.17, 95% CI: 1.04–1.28). Mothers who did not receive full antenatal care had higher odds of having LBW infants than those who received it (OR = 1.25, 95% CI: 1.25–1.35). Children born at public hospitals were less likely to be LBW (OR = 0.87, 95% CI: 0.77–0.83) than those born at home. In the second model, female children had higher odds of being LBW than male children (OR = 1.19, 95% CI: 1.16–1.22). The odds of being LBW were higher among first-borns than later-born. The odds ratio decreased significantly with increases in the mother's height. The

**Table 2. Estimated effects of maternal obstetric and socioeconomic factors on preterm birth in India, (2019–21).**

| Determinants | Model I | Model II | Model III | Model IV |
|---|---|---|---|---|
| **Maternal obstetric care** | | | | |
| **Full antenatal care** | | | | |
| Yes | Ref | | | Ref |
| No | 1.25***(1.20,1.29) | | | 1.32***(0.19, 1.45) |
| **Delivery complication** | | | | |
| No | Ref | | | Ref |
| Any complication | 1.08***(1.04,1.13) | | | 0.95 (0.87, 1.04) |
| **Place of delivery** | | | | |
| Home | Ref | | | |
| Public | 1.04 *(0.99, 1.09) | | | Ref |
| Private | 1.24***(1.17,1.31) | | | 1.19*** (1.06, 1.33) |
| **Previous caesarean delivery** | Ref | | | 1.46*** (1.26, 1.68) |
| No | | | | Ref |
| Yes | 1.10***(1.00,1.22) | | | 1.03(0.91, 1.15) |
| **Demographic Mogra** | | | | |
| **Mother's height** | | | | |
| <145 | | Ref | | Ref |
| 145–149 | | 1.03 (0.97,1.05) | | 1.03 (0.97,1.17) |
| 150–154 | | 1.02(0.97,1.07) | | 1.01(0.97,1.14) |
| >155 | | 1.04 (0.98,1.09) | | 0.97(0.98,1.11) |
| **Mother's age at delivery** | | | | |
| <24 | | Ref | | Ref |
| 24–29 | | 1.07 ***(1.02, 1.11) | | 0.98(0.87, 1.09) |
| 30 & above | | 1.01(0.97, 1.06) | | 0.88** (0.78, 0.98) |
| **Birth order** | | | | |
| 1 | | Ref | | Ref |
| 2 | | 1.03(0.98, 1.07) | | 1.03(0.98, 1.07) |
| 3 & above | | 1.01(0.97, 1.05) | | 1.01(0.97, 1.05) |
| **Birth interval** | | | | |
| <24 months | | Ref | | Ref |
| = >24 months | | 0.89***(0.86 0.93) | | 0.92***(0.87 0.96) |
| **Sex of child** | | | | |
| Male | | Ref | | Ref |
| Female | | 1.04 **(1.01, 1.07) | | 1.00 (0.93,1.08) |
| **Socio-economic status** | | | | |
| **Religion** | | | | |
| Hindu | | | 1.11 (1.07, 1.15) | 1.07 (0.97, 1.18) |
| Non-Hindu | | | Ref | Ref |
| **Wealth** | | | | |
| Poor | | | Ref | Ref |
| Middle | | | 1.03 ***(1.01, 1.10) | 0.99 (0.98, 1.10) |
| Rich | | | 1.16*** (1.11, 1.20) | 1.00 (0.88, 1.12) |
| **Mother's education** | | | | |
| Illiterate | | | Ref | Ref |
| Primary | | | 0.94**(0.85, 1.00) | 0.98(0.87, 1.11) |
| Secondary | | | 0.91***(0.96, 1.09) | 0.99(0.90, 1.09)) |
| Higher | | | 0.95*(1.05, 1.26) | 0.96(0.82, 1.12) |

*(Continued)*

**Table 2.** (Continued)

| Determinants | Model I | Model II | Model III | Model IV |
|---|---|---|---|---|
| **Place of residence** | | | | |
| Rural | | | | |
| Urban | | | 0.97 (0.93, 1.01) | 1.04 (0.93, 1.15) |
| **Source of drinking water** | | | | |
| Improved | | | Ref | Ref |
| Unimproved | | | 1.02 (0.97,1.07) | 1.01 (0.90,1.15) |
| **Sanitation facility** | | | | |
| Improved | | | Ref | Ref |
| Unimproved | | | 1.02 (0.97, 1.07) | 1.04 (0.95, 1.13) |
| **Region** | | | | |
| North | | | | Ref |
| Central | | | | 0.74***(0.67, 0.8) |
| East | | | | 0.70***(0.62,0.79) |
| North-East | | | | 0.55***(0.46, 0.65) |
| West | | | | 0.54***(0.46, 0.64) |
| South | | | | 0.67***(0.58,0.77) |

Note: Ref: Reference category;

*** p < 0.001,

** < 0.05,

* p < 0.1

odds of having LBW infants were inversely proportional to the mother's education and wealth quintile. The association of place of delivery, antenatal care, and history of previous caesarean section strengthened in Model IV after controlling for sociodemographic and economic factors. The North-Eastern part of India had the lowest odds of having LBW babies.

## Discussion

A growing body of literature documents LBW and PTB as major determinants of childhood morbidity, malnutrition, and premature mortality [20]. The long-term negative effects of PTB and LBW on child health can be a major obstacle in achieving India's SDG targets for child health, malnutrition, and general well-being. To our knowledge, this present study is the first in India to identify the correlates of adverse birth outcomes using nationally representative data and compare the associated factors of LBW and preterm birth. The central focus of our study was to address research gaps regarding macro-level estimates of PTB and LBW in India and understand how different correlates, such as obstetric factors, maternal health, demographic and socioeconomic characteristics, are associated with these two adverse birth outcomes. The salient findings of our study are as follows: First, the prevalence of PTB was higher in the northern states. Second, maternal obstetric and demographic factors, such as lack of antenatal care, history of previous caesarean delivery, complications during delivery, and low maternal height, were significantly associated with both preterm birth and LBW. Third, the results highlight a significantly different association between the child's sex and age at childbirth and socioeconomic correlates, such as maternal education and wealth status, with the two adverse birth outcomes. This calls for completely different policy approaches to address these two issues.

**Table 3. Estimated effects of maternal obstetric and socioeconomic factors on low birth weight in India, (2019–21).**

| Determinants | Model I | Model II | Model III | Model IV |
|---|---|---|---|---|
| | | Maternal obstetric care | | |
| **Full antenatal care** | | | | |
| Yes | Ref | | | Ref |
| No | 1.25***(1.25,1.35) | | | 1.29***(1.24,1.34) |
| **Pregnancy complication** | | | | |
| No | Ref | | | Ref |
| Any complication | 0.96(0.88, 1.03) | | | 1.13***(1.10, 1.16) |
| **Place of delivery** | | | | |
| Home | Ref | | | Ref |
| Public | 0.87**(0.77,0.98) | | | 0.83***(0.73,0.92) |
| Private | 1.01(0.98, 1.19) | | | 1.03(0.98, 1.19) |
| **Previous caesarean delivery** | | | | |
| No | Ref | | | Ref |
| Yes | 1.17***(1.04, 1.28) | | | 1.32***(1.19, 1.47) |
| | | Demographic | | |
| **Mother's height** | | | | |
| <145 | | Ref | | Ref |
| 145–149 | | 0.82** (0.78, 0.85) | | 0.88** (0.79, 0.99) |
| 150–154 | | 0.69*** (0.65, 0.71) | | 0.75*** (0.67, 0.84) |
| >155 | | 0.60*** (0.58, 0.63) | | 0.68*** (0.60, 0.76) |
| **Mother's age at delivery** | | | | |
| <24 | | Ref | | Ref |
| 24–29 | | 1.22*** (1.17, 1.27) | | 1.17*** (1.04, 1.30) |
| 30 & above | | 1.06**(1.01, 1.09) | | 1.01(0.91, 1.13) |
| **Birth order** | | | | |
| 1 | | Ref | | Ref |
| 2 | | 0.92***(0.89, 0.95) | | 0.95(0.83, 1.03) |
| 3 & above | | 0.99(0.96, 1.13) | | 1.05(0.96, 1.13) |
| **Birth interval** | | | | |
| <24 months | | Ref | | Ref |
| = >24 months | | 0.90***(0.86 0.93) | | 0.92***(0.88 0.95) |
| **Sex of child** | | | | |
| Male | | Ref | | Ref |
| Female | | 1.19*** (1.16, 1.22) | | 1.25*** (1.16, 1.33) |
| | | Socio-economic status | | |
| **Religion** | | | | |
| Hindu | | | 1.16**(1.12, 1.20) | 1.10**(1.00, 1.20) |
| Non-Hindu | | | | |
| **Wealth** | | | | |
| Rich | | | Ref | Ref |
| Middle | | | 1.26*** (1.20, 1.31) | 1.07 (0.96, 1.20) |
| Poor | | | 1.06 ***(1.03, 1.10) | 1.19 ***(0.96, 1.33) |
| **Mother's education** | | | | |
| Illiterate | | | Ref | Ref |
| Primary | | | 1.02 (0.98, 1.07) | 0.98 (0.88, 1.10) |
| Secondary | | | 0.92 ***(0.88, 0.95) | 0.85 ***(0.77, 0.93) |
| Higher | | | 0.72***(0.68 0.76) | 0.70***(0.60 0.81) |

*(Continued)*

**Table 3.** (Continued)

| Determinants | Model I | Model II | Model III | Model IV |
|---|---|---|---|---|
| **Place of residence** | | | | |
| Rural | | | Ref | Ref |
| Urban | | | 1.09*** (1.04, 1.12) | 1.20*** (1.09, 1.32) |
| **Source of drinking water** | | | | |
| Improved | | | Ref | Ref |
| Unimproved | | | 0.99 (0.94,1.04) | 1.02 (0.97,1.06) |
| **Sanitation facility** | | | | |
| Improved | | | Ref | Ref |
| Unimproved | | | 0.95 (0.92,0.98) | 1.01 (0.96,1.05) |
| **Region** | | | | |
| North | | | | Ref |
| Central | | | | 0.87**(0.79,0.97) |
| East | | | | 0.64***(0.57, 0.72) |
| North-East | | | | 0.43***(0.36, 0.50) |
| West | | | | 0.85**(0.74, 0.98) |
| South | | | | 0.62***(0.54, 0.71) |

Note: Ref: Reference category;

*** p < 0.001,

** < 0.05,

* p < 0.1

Factors such as a lack of health infrastructure, health professionals, and ANC check-ups are prevalent in most resource-poor states in India [40]. These states often have high rates of adverse child health outcomes, such as growth faltering and premature death. Addressing PTB and LBW among neonates could significantly improve child health outcomes in these states, enhancing public health and contributing to better human capital development. Given that these states are the most populous in India and are projected to contribute the most to the country's demographic dividend, such measures could have a significant impact [41, 42]. The study also highlights the high prevalence of PTB and LBW in the National Capital Region of Delhi. We speculate that high levels of air pollution in this region might contribute to these negative birth outcomes. Several studies have documented the negative effects of air pollution on birth outcomes, such as PTB and LBW, in China and India [43, 44].

The results of the multivariate regression analysis show that lack of full antenatal care increases the chances of a mother delivering PTB and LBW infants. The World Health Organization (WHO) recommends scaling up antenatal care to improve maternal and child health through five interventions: nutritional interventions, physical health check-ups, maternal and fetal assessment, preventive measures, and health system interventions [45]. ANC visits improve the mother's diet during pregnancy and encourage healthy weight gain [46, 47]. The results also highlight that anaemic mothers were more likely to deliver LBW babies. Iron deficiency among mothers could be a cause of insufficient fetal growth. Promoting optimal consumption of IFA tablets during ANC visits can improve the mother's overall nutritional status and bolster healthy fetal growth [48].

The history of previous caesarean delivery was significantly associated with adverse birth outcomes, but the biological reasons behind these associations remain unclear. One tentative reason might be the disruption of cervical integrity during the surgical procedure due to

factors such as cervical trauma, haemorrhage, bladder laceration, or late-stage caesarean section [49]. This damage can affect the function of the cervix and further increase the risk of PTB and LBW in future pregnancies [50]. Another study found that the weight of the placenta, low perfusion, and damage to the villi are all direct factors leading to adverse outcomes in the fetus [51]. Recent studies have suggested that the global surge in overall caesarean deliveries is mainly due to a rise in elective caesarean surgeries, which are usually supply-induced demand and have no strong medical evidence [52, 53]. Also, we observed that the poorer economic condition of the household is a significant predictor of LBW, which is similar to a hospital-based Indian study [54].

The results highlight the difference in the association of household wealth status with PTB and LBW. The odds of PTB were recorded to be higher among richer households. Richer mothers are often more informed about pregnancy complications and foetus health [55]. Medically unnecessary inductions and elective caesarean deliveries are also higher among these groups, leading to an increased risk of PTB [56]. The results also indicate that the odds of LBW were higher among poorer households. The study found a significant role in mothers' education in adverse birth outcomes. Uneducated mothers are often unable to access optimum antenatal care and proper nutrition during pregnancy, which deters fetal growth and predisposes LBW and PTB among newborns [56, 57]. Short maternal stature is an important indicator of adverse birth outcomes. It represents chronic malnutrition among mothers, which can lead to poor nutrition supply to the fetus during pregnancy [58].

The study found that children born in hospitals are more likely to be preterm, while home delivery was a significant factor in LBW. Due to improper growth in the fetal stages of their mothers, these infants have a higher chance of experiencing breathing problems, metabolic dysfunction, and apnoea of prematurity [4, 47, 59]. Thus, investment in Neonatal Intensive Care Units (NICUs) is necessary for premature babies, which are mostly available in private hospitals in India [60]. Moreover, caesarean delivery is common in cases of premature birth [49]. Therefore, most premature babies are reported in hospitals, specifically in private facilities. However, previous studies have found that mothers who deliver at home have a higher chance of having LBW infants, which is in line with the present study [27, 61].

The present study is based on nationally representative data that uses validated questionnaires and methodology [12]. However, our study has limitations that come with analyzing any sample survey data. Due to the cross-sectional nature of the data, the study is limited to associations of variables and not causality. Secondly, there is no objective measure or tool to test the reliability of self-reported data for the diagnosis of preterm birth. If LBW is not recorded on a card, the information on both preterm birth and LBW will depend on the interviewees' memory. This may increase the chances of recall bias.

Despite all these limitations, the study highlights various important factors that can improve India's birth outcomes and contribute to bettering child health indicators. The findings are indeed pivotal in providing inputs regarding the association of essential medical, demographic, social, and economic factors with PTB and LBW. The study indicates that the target populations for both are significantly different and require extensive policy approaches.

## Conclusions

The study found that there is an interstate disparity in adverse birth outcomes across the states of India. Although maternal obstetric factors were similarly associated with PTB and LBW, an in-depth research is required to understand the difference in the direction of their relationship with socioeconomic factors. The study of preterm birth at the population level is limited in India. This study could help in policy-making to reduce the burden of adverse birth outcomes.

The acceptance of antenatal care is increasing in all regions of India. This is a window of opportunity to reduce LBW and PTB through improved accessibility and availability of antenatal care. Educational interventions during antenatal visits could inform women about the adverse outcomes of unnecessary caesarean deliveries. Future studies should examine the effect of household food security and nutritional status during pre-pregnancy, which could explain all possible risk factors for adverse birth outcomes.

## Author Contributions

**Conceptualization:** Arup Jana.

**Data curation:** Arup Jana.

**Formal analysis:** Arup Jana.

**Methodology:** Arup Jana.

**Software:** Arup Jana.

**Visualization:** Arup Jana.

**Writing – original draft:** Arup Jana.

**Writing – review & editing:** Arup Jana.

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
