## [Decision Letter · Decision Letter 0]

12 Apr 2023

PONE-D-22-27840Correlates of Adverse Birth Outcomes in India: Evidence from National Family Health SurveyPLOS ONE

Dear Dr. Jana,

Thank you for submitting your manuscript to PLOS ONE. After careful consideration, we feel that it has merit but does not fully meet PLOS ONE’s publication criteria as it currently stands. Therefore, we invite you to submit a revised version of the manuscript that addresses the points raised during the review process.

ACADEMIC EDITOR: Be sure to:Clarify the focus of the analysis; general birth outcomes or PTB, LBW? this should reflect in the titleRe-organise presentation of results.Address ALL comments from reviewers.==============================

We look forward to receiving your revised manuscript.

Kind regards,

Benedict Weobong, Ph.D

Academic Editor

PLOS ONE

“No”

“No”

5. We note that [Figure 1] in your submission contain [map/satellite] images which may be copyrighted. All PLOS content is published under the Creative Commons Attribution License (CC BY 4.0), which means that the manuscript, images, and Supporting Information files will be freely available online, and any third party is permitted to access, download, copy, distribute, and use these materials in any way, even commercially, with proper attribution. For these reasons, we cannot publish previously copyrighted maps or satellite images created using proprietary data, such as Google software (Google Maps, Street View, and Earth). For more information, see our copyright guidelines: http://journals.plos.org/plosone/s/licenses-and-copyright.

a. You may seek permission from the original copyright holder of Figure(s) [#] to publish the content specifically under the CC BY 4.0 license. 

Reviewers' comments:

Reviewer's Responses to Questions

**Comments to the Author**

1. Is the manuscript technically sound, and do the data support the conclusions?

Reviewer #1: Yes

Reviewer #2: Partly

2. Has the statistical analysis been performed appropriately and rigorously? 

Reviewer #1: Yes

Reviewer #2: I Don't Know

3. Have the authors made all data underlying the findings in their manuscript fully available?

Reviewer #1: Yes

Reviewer #2: No

4. Is the manuscript presented in an intelligible fashion and written in standard English?

Reviewer #1: Yes

Reviewer #2: Yes

5. Review Comments to the Author

Reviewer #1: Also, attached separatelyAbstract

The abstract is well-written, overall. However, the results section of the abstract requires some further clarification. For example, the author stated that “However, a few correlates were found to be differently associated with PTB and LBW. Mothers who belonged to richer wealth status had a higher risk of PTB, the relationship was reversed in the case of LBW” The authors fell short of reporting those few correlates. It would be instructive to report those different correlates.

Introduction

the current introduction section looks great, but I was expecting to see an overview of the prevalent adverse birth outcomes globally before the focus on the most common/prevalent ones. From the onset, the author explicitly identified PTB and LBW as the key adverse birth outcomes and missed the point about ensuring the readers have a sense of the others before narrowing on these two. If the author intends to proceed, then a title will need to be modified to reflect PTB and LBW. At least, the introduction section should provide the reader with sufficient information about the various adverse birth outcomes and some of the reported predictors.

Also, a see a bit of dissonance in reporting studies on general birth outcomes on one hand, and PTB and LBW on the other hand. I suggest a presentation of the literature first on adverse birth outcomes before a focus on the two most prevalent birth outcomes.

Is it possible to compare the state of the evidence on adverse birth outcomes globally and what pertains to India? Are there any differences or uniqueness between Indian and other countries?

Methods and Results

It will be important for the author to report the sampling approach adopted in the national survey from which the data was also sampled or extracted for this study. Also, was permission sought before the data was accessed or the data is freely accessible to the public? Again, it is clear from the methods and presentation of the results that the focus of the paper is PTB and LBW and so I suggest the title be modified to reflect this. I believe both bivariate and multivariable logistic regression analyses were carried out, so it is important to emphasize that.

Overall, the results as presented are okay but can I suggest the author reorganizes the results into two sections? The first can take the form of descriptive results where data on the prevalence and risk factors are presented. The second will be logistics regression which will present results including from models 1-4 as seen in table 2.

Discussion

There are some grammatical errors that need to be attended to via thorough proofreading. For example, see line 299. I believe the author meant to use the word “off-track”.

Reviewer #2: Because the data link http://rchiips.org/NFHS/NFHS-4Report.shtml is not available (HTTP Error 503. The service is unavailable) it was difficult to review the paper. The following feedback is based on the following original data set, assuming it is the correct one: https://dhsprogram.com/methodology/survey/survey-display-541.cfm and the principle concern is that the data set does not clearly differentiate PTB from LBW and it is unclear how the author has done this. In addition, substantial conclusions are drawn from LBW data even though it is described as a weaker dataset to be used with caution in the original data set. The other key query is why birth interval is not considered. If tested and found not to be relevant, this should be explained.

The remaining feedback is based on the paper alone, not refering to the broader / original data set.

Ethics: in at the start of the submission, reference is made to written informed consent, whereas the rest of the paper refers to use of national anonymised data only, please clarify.

Rows 59-61 the clarity of the statement can be reviewed

Rows 170-192 it would be helpful to know whether any of these differences are statistically significant

Rows 265-272 other studies have also outlined the linkages with CS and could be useful to look at to support the hypothesis, eg https://www.ncbi.nlm.nih.gov/pmc/articles/PMC8350719/#:~:text=Pregnant%20women%20who%20had%20previous,disease%20as%20well%20as%20gestational (from google)

Rows 287-289 check wording

6. PLOS authors have the option to publish the peer review history of their article (what does this mean?). If published, this will include your full peer review and any attached files.

Reviewer #1: No

Reviewer #2: No

---

## [Author Response · Author response to Decision Letter 0]

15 May 2023

REPLY TO REVIEWERS’ COMMENTS

Submission ID: PONE-D-22-27840 

Title: Correlates of Adverse Birth Outcomes in India: Evidence from National Family Health Survey 

I sincerely thank the editor and the anonymous reviewers for their appreciation, thoughtful

suggestions and comments. I have revised the manuscript as per the suggestions and

comments of the reviewers. A point-by-point reply to each of the reviewer’s comment is given below.

Editors comments:

We note that [Figure 1] in your submission contain [map/satellite] images which may be copyrighted. All PLOS content is published under the Creative Commons Attribution License (CC BY 4.0), which means that the manuscript, images, and Supporting Information files will be freely available online, and any third party is permitted to access, download, copy, distribute, and use these materials in any way, even commercially, with proper attribution. For these reasons, we cannot publish previously copyrighted maps or satellite images created using proprietary data, such as Google software (Google Maps, Street View, and Earth). For more information, see our copyright guidelines: http://journals.plos.org/plosone/s/licenses-and-copyright.

Response: The maps were prepared by the author using the dataset (NFHS), that is mentioned in the figure. The shapefile used in the study was provided by the NFHS. No figures, maps, or images from third parties have been used in the study. 

Reviewer's comments:

Reviewer #1: 

Abstract

The abstract is well-written, overall. However, the results section of the abstract requires some further clarification. For example, the author stated that “However, a few correlates were found to be differently associated with PTB and LBW. Mothers who belonged to richer wealth status had a higher risk of PTB, the relationship was reversed in the case of LBW” The authors fell short of reporting those few correlates. It would be instructive to report those different correlates.

Response: Thank you for the comment. The results have been modified accordingly in the abstract, and more information has been added about the same.

“Mothers belonging to richer wealth status had higher chances of having preterm birth (OR=1.16, 95% CI: 1.11-1.20) in comparison to poor mothers. While, the odds of having LBW infants were found to be increased with the decreasing level of the mother's education and wealth quintile.”

Introduction

the current introduction section looks great, but I was expecting to see an overview of the prevalent adverse birth outcomes globally before the focus on the most common/prevalent ones. From the onset, the author explicitly identified PTB and LBW as the key adverse birth outcomes and missed the point about ensuring the readers have a sense of the others before narrowing on these two. If the author intends to proceed, then a title will need to be modified to reflect PTB and LBW. At least, the introduction section should provide the reader with sufficient information about the various adverse birth outcomes and some of the reported predictors.

Also, a see a bit of dissonance in reporting studies on general birth outcomes on one hand, and PTB and LBW on the other hand. I suggest a presentation of the literature first on adverse birth outcomes before a focus on the two most prevalent birth outcomes.

Is it possible to compare the state of the evidence on adverse birth outcomes globally and what pertains to India? Are there any differences or uniqueness between Indian and other countries?

Response: Thank you. The introduction of the manuscript has been modified according to your suggestion. 

“The adverse birth outcome (ABO)is a complex outcome influenced by multiple factors, encompassing preterm birth, low birth weight, stillbirth, macrosomia, and congenital anomalies. Preterm birth (PTB) is defined as delivery before 37 weeks of gestation, while low birth weight (LBW) refers to infants born weighing less than 2500 g [1,2]. Stillbirth refers to the delivery of a baby without any signs of life, while macrosomia describes newborns who are larger than the average size. Congenital anomalies are a diverse range of structural or functional abnormalities present at birth, originating during prenatal development [3]. Globally, there are approximately 15 million premature births [4] and over 20 million infants born with LBW each year [5]. The prevalence of PTB worldwide is 10.6%, with South Asia accounting for more than one-third of the burden. Additionally, nearly 3 million stillbirths occur annually worldwide, with 98% of them happening in developing countries [6].”

A global scenario and details of adverse birth outcomes have been added in the revision.

“Preterm birth is a significant public health challenge globally, with PTB being the leading cause of neonatal deaths and the second leading cause of under-five mortality [7]. The Sustainable Development Goals (SDGs) set out to reduce preventable deaths in newborns and children under the age of five by 2030, with a goal of reducing neonatal mortality to at least 12 per 1,000 live births and under-five mortality to at least 25 per 1,000 live births [8]. However, in India, PTB remains a pressing issue, with an estimated 3.5 million cases of PTB in 2010, accounting for 23.4% of the global burden [9].”

The title has been changed to “Correlates of Adverse Birth Outcomes in India: A Special Focus on Low Birth Weight and Preterm Birth”. 

More information about the adverse birth outcomes and their predictors has been added in the introduction. First highlighted the predictors of adverse birth outcomes then specific factors of LBW and PTB. 

“Adverse birth outcomes are influenced by a variety of determinants. These can include maternal factors, such as maternal age, race/ethnicity, and preexisting health conditions (e.g., diabetes, hypertension) [20,21]; lifestyle factors, such as tobacco, alcohol, and drug use during pregnancy; inadequate prenatal care; and social determinants of health, such as socioeconomic status, education level, access to healthcare, and environmental factors (e.g., exposure to pollution, neighborhood safety) [22,23]. Additionally, genetics, maternal stress, and intergenerational factors can also play a role in determining birth outcomes [24]. A complex interplay of these determinants can impact fetal development and increase the risk of adverse birth outcomes, highlighting the importance of addressing multifactorial influences to improve maternal and infant health outcomes.”

The prevalence of LBW and PTB in India and its neighbouring countries have been added in the introduction of the revised manuscript. 

 “The prevalence of LBW and PTB in India, at 18% and 13%, respectively, is higher than in neighbouring countries such as Sri Lanka (LBW:15.9 and PTB: 7.0), Nepal (LBW:12.3 and PTB: 5.3), Myanmar (LBW:8.1 and PTB: 10.4), and China (LBW:6.9 and PTB: 0) [14]. These findings emphasize the urgent need for improved interventions and strategies to address the prevalence of PTB and LBW in India and prevent neonatal deaths and under-five mortality.”

Methods and Results

It will be important for the author to report the sampling approach adopted in the national survey from which the data was also sampled or extracted for this study. Also, was permission sought before the data was accessed or the data is freely accessible to the public? Again, it is clear from the methods and presentation of the results that the focus of the paper is PTB and LBW and so I suggest the title be modified to reflect this. I believe both bivariate and multivariable logistic regression analyses were carried out, so it is important to emphasize that.

Overall, the results as presented are okay but can I suggest the author reorganizes the results into two sections? The first can take the form of descriptive results where data on the prevalence and risk factors are presented. The second will be logistics regression which will present results including from models 1-4 as seen in table 2. 

Response: the sample details such as the sampling frame, sampling technique and approach have been added in the Data section of the manuscript. 

“The NFHS-5 sample was designed to provide accurate estimates of all key indicators at the national, state, and district levels (for all 707 districts in India as of March 31, 2017). To achieve this, a sample size of around 636,699 households was determined. The primary sampling units (PSUs) were villages, selected with probability proportionate to size, and the rural sample was selected using a two-stage sample design. In the second stage, 22 households were chosen at random from each PSU. In urban areas, a two-stage sample design was also used, with 22 households selected in each Census Enumeration Block (CEB) in the second stage. After mapping and listing households in the selected first-stage units, households were selected for the second stage in both urban and rural areas. The survey covered 724,115 women in the reproductive age group of 15-49 years, and information about 232,920 children was collected from their mothers. Of these children, a total of 170,253 most recent (i.e., last-born) births were included in the study.”

The manuscript now includes the necessary information about ethical approval and consent. 

“The study is based on publicly available secondary data. It is certified that all applicable institutional and governmental regulations concerning the ethical use of human volunteers were followed during the course of the survey. Verbal and written informed consent was obtained from all participants or their parent/legal guardian if they were not of mature age or below 18 years old.”

As per your suggestion, the results of LBW and PTB have ben divided into two section to interpret the bivariate and multivariate logistic regression results. The title has been modified. The bivariate and multivariate have been used in the study, added in the statistical analysis section. Presenting all the details of the results of logistic regression will be too lengthy thus most important results are mentioned only. 

Discussion

There are some grammatical errors that need to be attended to via thorough proofreading. For example, see line 299. I believe the author meant to use the word “off-track”.

Response: The grammatical errors have been removed from the manuscript. Hope you will be satisfied with the proofreading. 

Reviewer #2: 

Because the data link http://rchiips.org/NFHS/NFHS-4Report.shtml is not available (HTTP Error 503. The service is unavailable) it was difficult to review the paper. The following feedback is based on the following original data set, assuming it is the correct one: https://dhsprogram.com/methodology/survey/survey-display-541.cfm

Response: Both the links are correct. NFHS is another name of Indian DHS. Thank you for attaching the DHS link. The data availability link has been edited as per your suggestion. 

“Availability of data and materials: The study utilized the secondary data, which is publicly available through https://dhsprogram.com/methodology/survey/survey-display-541.cfm”

the principle concern is that the data set does not clearly differentiate PTB from LBW and it is unclear how the author has done this.

Response: Thank you for pointing this out. All the details are mentioned in the data and methods section of the manuscript. The questions have been asked in the survey and methods to estimate LBW and PTB are mentioned in the manuscript. You will find the information related to LBW in the report, but the PTB is not available. 

“Preterm birth is defined as a live birth before 37 completed weeks of gestation and low birth

weight is defined by the World Health Organization (WHO) as weight at birth less than 2500

grams [36]. Jana et al., (2022) was followed in this study to measure preterm birth using the calendar method of DHS [37]. The NFHS collected data on birth weight using the following questions: Was (name of the child) weighed at birth? How much did (name of the child) weigh? The information was reported in two ways; first, the mother's recall about her baby weight, and second, reported with the help of any card of their baby weight [12]. The other outcome variable was preterm birth which was estimated using the duration of pregnancy. The outcome variables are dichotomous 0 signifying child did not have LBW/ was not preterm and 1 denoting child had LBW/ was preterm.”

 In addition, substantial conclusions are drawn from LBW data even though it is described as a weaker dataset to be used with caution in the original data set. 

Response: That is the only dataset available represents the national level estimates. More details about the LBW in the datasets and it is weaker or not that can be found from my another study: Jana, A., Saha, U. R., Reshmi, R. S., & Muhammad, T. (2023). Relationship between low birth weight and infant mortality: evidence from National Family Health Survey 2019-21, India. Archives of Public Health, 81(1), 1-14.

Here we have proved that 8% missing data of birth weight that was not measured after birth in India, does not effect the outcomes. 

The other key query is why birth interval is not considered. If tested and found not to be relevant, this should be explained.

Response: The variable has been added in the analysis.

The remaining feedback is based on the paper alone, not refering to the broader / original data set.

Ethics: in at the start of the submission, reference is made to written informed consent, whereas the rest of the paper refers to use of national anonymised data only, please clarify. 

Response: The details of ethical approval has been added in the data section of the manuscript.

“The study is based on publicly available secondary data. It is certified that all applicable institutional and governmental regulations concerning the ethical use of human volunteers were followed during the course of the survey. Verbal and written informed consent was obtained from all participants or their parent/legal guardian if they were not of mature age or below 18 years old.”

Rows 59-61 the clarity of the statement can be reviewed

Response: The statements have been modified in the revised manuscript. 

Rows 170-192 it would be helpful to know whether any of these differences are statistically significant

Response: Thank you for the comment. The chi squired value has been added in the table 1 that represents the significance level. 

Rows 265-272 other studies have also outlined the linkages with CS and could be useful to look at to support the hypothesis, eg https://www.ncbi.nlm.nih.gov/pmc/articles/PMC8350719/#:~:text=Pregnant%20women%20who%20had%20previous,disease%20as%20well%20as%20gestational (from google)

Response: Thank you for providing the article. The paragraph edited an cited the reference. 

“The history of previous caesarean delivery was significantly associated with adverse birth outcomes, but the biological reasons behind these associations remain unclear. One tentative reason might be the disruption of cervical integrity during the surgical procedure due to factors such as cervical trauma, hemorrhage, bladder laceration, or late-stage caesarean section [49]. This damage can affect the function of the cervix and further increase the risk of PTB and LBW in future pregnancies [50]. Another study found that the weight of the placenta, low perfusion, and damage to villi are all direct factors leading to adverse outcomes in the fetus [51]. Recent studies have suggested that the global surge in overall caesarean deliveries is mainly due to a rise in elective caesarean surgeries, which are usually supply-induced demand and have no strong medical evidence [52,53]. Also, we observed that the poorer economic condition of the household is a significant predictor of LBW, which is similar to a hospital-based Indian study [54].”

Rows 287-289 check wording

Response: Edited.

---

## [Editor Report · Decision Letter 1]

22 May 2023

PONE-D-22-27840R1Correlates of Adverse Birth Outcomes in India: A Special Focus on Low Birth Weight and Preterm BirthPLOS ONE

Dear Dr. Jana,

Thank you for submitting your manuscript to PLOS ONE. After careful consideration, we feel that it has merit but does not fully meet PLOS ONE’s publication criteria as it currently stands. Therefore, we invite you to submit a revised version of the manuscript that addresses the points raised during the review process.Title: we note you've agreed with the reviewers and have now focussed your analysis on LBW and PTB. Based on this, we think your title can be simplified. e.g. "Correlates of Low Birth Weight and Pre-term Birth in India"Abstract line 30: check use of word 'determinates'. This is a typo and should be corrected. Please do a  thorough editorial review of the entire manuscript to ensure these issues are avoided.Abstract lines 33-35: you're still maintaining the broader term 'adverse birth outcomes'- this should be corrected.Please submit your revised manuscript by Jul 06 2023 11:59PM. If you will need more time than this to complete your revisions, please reply to this message or contact the journal office at plosone@plos.org. Please include the following items when submitting your revised manuscript:A rebuttal letter that responds to each point raised by the academic editor and reviewer(s). You should upload this letter as a separate file labeled 'Response to Reviewers'.A marked-up copy of your manuscript that highlights changes made to the original version. You should upload this as a separate file labeled 'Revised Manuscript with Track Changes'.An unmarked version of your revised paper without tracked changes. You should upload this as a separate file labeled 'Manuscript'.If applicable, we recommend that you deposit your laboratory protocols in protocols.io to enhance the reproducibility of your results. Protocols.io assigns your protocol its own identifier (DOI) so that it can be cited independently in the future. For instructions see: https://journals.plos.org/plosone/s/submission-guidelines#loc-laboratory-protocols. Additionally, PLOS ONE offers an option for publishing peer-reviewed Lab Protocol articles, which describe protocols hosted on protocols.io. Read more information on sharing protocols at https://plos.org/protocols?utm_medium=editorial-email&utm_source=authorletters&utm_campaign=protocols.

We look forward to receiving your revised manuscript.

Kind regards,

Benedict Weobong, Ph.D

Academic Editor

PLOS ONE
---

## [Author Response · Author response to Decision Letter 1]

23 May 2023

REPLY TO REVIEWERS’ COMMENTS

Submission ID: PONE-D-22-27840 

Title: Correlates of Low Birth Weight and Preterm Birth in India

I sincerely thank the editor and the anonymous reviewers for their appreciation, thoughtful

suggestions and comments. I have revised the manuscript as per the suggestions and

comments of the reviewers. A point-by-point reply to each of the reviewer’s comment is given below.

Reviewer's comments:

Title: we note you've agreed with the reviewers and have now focussed your analysis on LBW and PTB. Based on this, we think your title can be simplified. e.g. "Correlates of Low Birth Weight and Pre-term Birth in India"

Response: Thank you for suggesting the title. I have modified the title as per your suggestion. 

Abstract line 30: check use of word 'determinates'. This is a typo and should be corrected. Please do a thorough editorial review of the entire manuscript to ensure these issues are avoided.

Response: Thank you for the precise observation. I have revised the manuscript and removed all the typo mistakes. 

Abstract lines 33-35: you're still maintaining the broader term 'adverse birth outcomes'- this should be corrected.

Response: Modified as per your suggestion.

---

## [Editor Report · Decision Letter 2]

15 Jun 2023

Correlates of Low Birth Weight and Preterm Birth in India

PONE-D-22-27840R2

Dear Dr. Jana,

We’re pleased to inform you that your manuscript has been judged scientifically suitable for publication and will be formally accepted for publication once it meets all outstanding technical requirements.

Kind regards,

Benedict Weobong, Ph.D

Academic Editor

PLOS ONE
---

## [Editor Report · Acceptance letter]

8 Aug 2023

PONE-D-22-27840R2 

Correlates of Low Birth Weight and Preterm Birth in India 

Dear Dr. Jana:

I'm pleased to inform you that your manuscript has been deemed suitable for publication in PLOS ONE. Congratulations! Your manuscript is now with our production department. 

Kind regards, 

on behalf of

Dr. Benedict Weobong 

Academic Editor

PLOS ONE